# Effect of Dietary Ensiled Olive Cake Supplementation on Performance and Meat Quality of Apulo-Calabrese Pigs

**DOI:** 10.3390/ani13122022

**Published:** 2023-06-18

**Authors:** Pasquale Caparra, Luigi Chies, Manuel Scerra, Francesco Foti, Matteo Bognanno, Caterina Cilione, Paolo De Caria, Salvatore Claps, Giulia Francesca Cifuni

**Affiliations:** 1Division of Animal Production, Department of Agriculture, Mediterranean University of Reggio Calabria, Via dell’Università, 25, 89124 Reggio Calabria, Italy; lchies@unirc.it (L.C.); manuel.scerra@unirc.it (M.S.); francesco.foti@unirc.it (F.F.); matteo.bognanno@unirc.it (M.B.); caterina.cilione@unirc.it (C.C.); paolo.decaria@unirc.it (P.D.C.); 2Council for Agricultural Research and Economics—Research Centre for Animal Production and Aquaculture, S.S.7 Via Appia, 85051 Bella Muro, Italy; salvatore.claps@crea.gov.it (S.C.); giuliafrancesca.cifuni@crea.gov.it (G.F.C.)

**Keywords:** circular economy, olive by-product, fatty acid composition, silage olive cake, fattening pig

## Abstract

**Simple Summary:**

A valid strategy for increasing the sustainability of food production systems is the use of industrial by-products in livestock diets, which allows food processing industries to profitably mitigate the costs generated by waste disposal. The agro-industrial olive sector produces large quantities of olive by-products, considered extremely toxic, with a highly adverse environmental impact. Thus, to valorise a by-product of the oil industry, the aim of this study was to evaluate the inclusion of different levels of ensiled olive cake in the fattening diets of Apulo-Calabrese pigs as a strategy to partially replace the conventional cereal-based diet and improve animal performance and meat fatty acid profile. The results showed that the inclusion of ensiled olive cake in the diet improves the fatty acid composition of meat, improving health by increasing the oleic acid level (C8:1n-9, *p* < 0.001), decreasing the linoleic acid level (C18:2n-6, *p* < 0.001) and altering the polyunsaturated/saturated ratio (P/S, *p* < 0.001). In conclusion, ensiled olive cake is suitable for dietary supplementation at up to 40% for fattening pigs and could represent a valid strategy for naturally improving the nutritional value of meat and valorising a by-product of the olive industry.

**Abstract:**

The aim of this study was to evaluate the inclusion of different amounts of ensiled olive cake, a major pollutant from olive oil production, in the fattening diets of 30 Apulo-Calabrese pigs as a strategy to partially substitute the traditional cereal-based diet and improve animal performance and meat fatty acid composition. The animals, during a fattening period of 120 days, were fed with three dietary treatments containing increasing levels of ensiled olive cake: 0% (C), 20% (OC20) and 40% (OC40) on dry matter. No effect of the dietary treatment was found on the animal performance and proximate meat analysis results. The inclusion of ensiled olive cake in the diet led to differences in the fatty acid (FA) profile of intramuscular fat, with a higher proportion of monounsaturated fatty acid (MUFA; *p* < 0.001) and oleic acid (C8:1n-9, *p* < 0.001) and a lower concentration of polyunsaturated (PUFA, *p* < 0.001) and linoleic acid (C18:2n-6, *p* < 0.001). In conclusion, the supplementation of ensiled olive cake at up to 40% in the diets of fattening pigs could represent a useful strategy in Mediterranean areas to naturally improve the nutritional value of meat and valorise a by-product of the olive industry, reducing its environmental impact and promoting the exploitation of this local feed resource according to the principles of the circular economy.

## 1. Introduction

Recently, due to the global intensification of food production, which continues to generate considerable quantities of co-products and food waste, there has been considerable social and environmental pressure for the effective re-use of agricultural industry residues. A valid strategy to increase the sustainability of food production systems is the use of industrial by-products in livestock diets, which allows food processing industries to profitably mitigate the costs generated by waste disposal [1].

Furthermore, in accordance with a circular economy, one of the key challenges to preserving natural resources and ensuring environmental sustainability is the use of agro-industrial by-products as replacements for traditional feedstuffs or to improve animal products [2].

The inclusion of agricultural by-products in animal nutrition could have significant economic benefits by reducing the costs of animal feeding and improving the chemical, physical and sensory characteristics of animal products [3,4] due to their high content of bioactive secondary compounds. In Italy, 1,156,344 hectares of land are used for the cultivation of olive trees, with an average annual production of 338 thousand tons of oil, of which about 70% is produced in Southern Italy [5]. The agro-industrial olive sector produces, in addition to olives and olive oil, large quantities of olive by-products. These by-products are considered extremely toxic, with a high adverse environmental impact.

The chemical properties of olive cake are affected by the type of olive, the amounts of its constituents (skin, pulp and stone) and the oil extraction process [6]. The high levels of residual oil and oleic acid in olive cake make it suitable for use in animal feed. Positive results have been obtained in pigs [7,8], cattle [9], small ruminants [10,11], broilers [12] and rabbits [13]. Incorporating olive cake into the pig diet may improve their growth performance, reduce carcass fat thickness [14] or provide more energy to ruminants [15]. The high lignin content associated with the presence of stone, on the other hand, may limit its use in piglets unless it is completely removed [16].

In finishing pigs, many authors [8,17,18] found that increasing the level of dietary olive cake supplementation results in a decrease in saturated fatty acid (SFA) and an increase in monosaturated fatty acid (MUFA), with an improved meat fatty acid composition in relation to human health. Furthermore, supplementing the diet of finishing pigs with partially defatted olive cake (120 g/kg) results in an increase in MUFA concentration in meat without affecting the growth performance, carcass quality or microbial counts [14].

Furthermore, olive by-products are available seasonally, and their use as animal feed throughout the year necessitates preservation and storage. The main issue with preserving olive cake is that it contains a high percentage of water and oil, and long-term storage may result in mould formation and nutrient losses [19]. The silage preservation method is a simple and effective way to preserve olive cake while increasing its nutritional value [20]. Thus, to valorise a by-product of the oil industry, the aim of this study was to evaluate the inclusion of different levels of ensiled olive cake in the fattening diets of Apulo-Calabrese pigs as a strategy to partially replace the conventional cereal-based diet and improve animal performance and meat fatty acid profile.

## 2. Materials and Methods

### 2.1. Ethical Statement

In this trial, the handling and experimental procedures were carried out according to international guidelines [21] and approved by the Animal Welfare Committee of the Mediterranean University of Reggio Calabria (Prot. No. 3015).

### 2.2. Diets, Animals and Experimental Design

The present trial was conducted on a private farm located in the Calabria region (Southern Italy), mainly focused on the production of heavy Apulo-Calabrese pigs. The Apulo-Calabrese pig, also known as the Nero Calabrese, is an Italian autochthonous breed from the Calabria region, reared traditionally in a free-range system that is based on natural woodland resources (acorns, chestnuts, berries, tubers, roots and pasture).

However, due to the seasonality of natural woodland products (which are only abundant in autumn and winter), the traditional pig-rearing system has some limitations. Thus, both the maintenance of traditional outdoor systems and the development of semi-intensive farming have been suggested to improve the breed’s profitability [22].

The experimental trials were carried out on a private farm whose traditional free-range rearing system for Apulo-Calabrian pigs includes a fattening period based on concentrate feeds complemented with free-range rearing.

The study was performed, during the fattening period (120 days), on 30 castrated male pigs of the Apulo-Calabrese breed that were randomly allotted to three groups in accordance with dietary treatments and balanced to their age and body weight (average initial body weight of 84.83 ± 3.52 kg).

In the daytime, all pigs were kept outdoors for about six to seven hours in a fenced wooded area of approximately two hectares and fed on natural woodland products (acorns, chestnuts, berries, tubers, roots and pasture). The fenced wooded area of grazing was divided into two paddocks (one hectare each) so that animals grazed rotationally. In the evening, when the animals returned from the pasture, they were placed in individual 4 m^2^ cages and fed with the experimental dietary treatments for 120 days of fattening.

The pigs were fed ad libitum (with an average consumption of 3.0 kg per head per day on fresh weight) with three dietary treatments (Table 1) formulated to contain: only concentrate (C; 10 pigs), concentrate with the inclusion of olive cake silage at 20% of the dry matter weight (OC20; 10 pigs) and concentrate with the inclusion of olive cake silage at 40% of the dry matter weight (OC40; 10 pigs). The control diet was formulated using conventional ingredients (mainly barley and maize grains and wheat bran) locally produced and traditionally employed in the Calabria region for pig feeding. Furthermore, to preserve the natural quality attributes of this environmentally sustainable pig-farming system, which is highly appreciated by consumers, for our research, we choose to source organic olive cake from olive farms with organic certifications. The diets containing ensiled olive cake (OC20 and OC40) were fed in the form of total mixed rations. All diets were designed to be isoproteic and isoenergetic (Table 1). Every day, before the addition of fresh feed, the amounts of feed offered and refused were recorded and the dry matter content was determined to calculate individual voluntary feed intakes (DMI). From the start to the end of the experiment, pigs were weighed every 15 days to determine the average daily gain (ADG) and feed conversion ratio (FCR).

The fresh olive cake was obtained from a two-phase continuous centrifugation system for olive oil extraction (Pieralisi MAIP Group S.p.A., Jesi, Italy). After the oil extraction, the virgin olive cake was stoned, immediately accumulated and well compacted in the silo, which was covered with a black polyethene film and firmly closed. The olive cake was kept in the silo for about three months before use and stored during the experimental trials. No additives were added to olive cake silages.

### 2.3. Chemical Analyses of Experimental Diets

The diet samples, relative to the dietary treatments, were collected daily and pooled weekly, followed by storage at −20 °C until further analysis. Table 1 shows the chemical composition of experimental diets. The dry matter (DM), ether extract (EE) and ash contents of diets were performed according to AOAC [23] procedures. Fibre fractions, including NDF, ADF and ADL, were measured according to the method described by Van Soest et al. [24]. The method of Gray et al. [25] was used to analyse the fatty acid composition of dietary treatments.

### 2.4. Slaughter, Carcass Characteristics and Meat Sampling

All pigs were slaughtered on the same day at a commercial slaughterhouse with an average weight of 145.24 ± 7.75 kg. All animals were treated and slaughtered according to the welfare regulations and the EU Council Regulation [26]. Following slaughter, the carcasses were moved to a 4 °C cooling chamber for 24 h.

Successively, the carcass weight was recorded, and the dressing percentage expressed was calculated. A section of 300 g (± 30 g) of loin (*Longissimus thoracis* muscle, LT) was excised at the level of 10th/14th thoracic vertebrae from each carcass and was vacuum sealed immediately and refrigerated during transport to the laboratory. Then, 200 g of each muscle was vacuum sealed and used after 24 h of storage at ±4 °C for the measurement of the physical and chemical characteristics. For fatty acid analysis, 100 g of each sample was vacuum packed and stored at −20 °C until subsequent analysis.

### 2.5. Colour, Physical and Proximate Analyses for Meat Samples

The colour parameters were measured after blooming (30 min), using a MINOLTA CR300 spectrophotometer with D65 illuminance, with an 11 mm aperture and 10° observer angle. The parameters L*, a* and b*, representing lightness, redness and yellowness, were measured four times on the cut surface and averaged.

The pH was measured at 48 h post-mortem on LT with a portable pH meter (Hanna HI982) equipped with automatic temperature compensation. The pH was measured using calibrated pH standard buffers at pH 4.0 and pH 7.0.

The dry matter and proximate composition of meat samples were determined in duplicate using AOAC standard procedures [23].

The protein determination was performed using the Kjeldahl method, multiplying the nitrogen content by 6.25. The fat content was extracted by Soxhlet apparatus on meat samples hydrolysed by HCl 3N. The moisture was calculated on 10–15 g pieces of the meat samples, which were dried in the oven at C 105 °C overnight. The muffle furnace at 550 ± 1 °C was used for the determination of the ash content.

### 2.6. Fatty Acids Analysis

The lipids were extracted, in duplicate, from 5 g of meat samples according to the procedure of Folch et al. [27]. Aliquots of 100 mg of lipid, with nonanoic acid methyl ester (20 mg/mL; standard no. 245895, Sigma-Aldrich, St。 Louis, MO, USA) as the internal standard, were methylated by adding 1 mL of hexane and 0.05 mL of 2 N methanolic potassium hydroxide [28].

Gas chromatography analysis was performed using a GC 6890N (Agilent, Inc., Santa Clara, CA, USA) instrument connected to a 60 m-length fused silica capillary column coated with 100% cyanopropyl polysiloxane (DB 23, Aligent J&W, Santa Clara, CA, USA) and with an internal diameter of 0.25 mm and film thickness of 0.25 μm. The operating conditions were as follows: a helium flow rate of 1.2 mL/min, a flame ionisation detector at 250 °C, a split–splitless injector at 230 °C with a split ratio of 1:100, and an injection volume of 1 μL. The temperature program of the column was: 5 min at 60 °C increased at a rate of 14 °C/min to 165 °C. A subsequent increase of 2 °C/min led to a final temperature of 225 °C for 20 min [29]. To identify the individual fatty acids, mixtures of standard fatty acids (Supelco FAME mix37 47885U, Sigma-Aldrich, St Louis, MO, USA) were used. Fatty acids were expressed as mg/100 gr of muscle. The atherogenic index (AI) and thrombogenic index (TI) were determined in accordance with Ulbricht and Southgate [30], while the hypocholesterolemic/hypercholesterolemic ratio was calculated as reported by Bialek et al. [31].

### 2.7. Statistical Analysis

The Sas procedure [32] was performed for data analysis, using the dietary treatments as factors in ANOVA analysis. The mean values were compared by Fisher’s LSD test, and the value of *p* < 0.05 was considered as the significant difference level. To determine the relationship between the different variables and detect the most important factors of variability, standard PCA was applied to the fatty acid composition of meat samples.

In the PCA model, only the fatty acids showing significant differences (*p* < 0.05) were considered to be variables.

## 3. Results

### 3.1. Animal Performance and Meat Quality Measurements

As shown in Table 2, no effect of the dietary treatment was found on the final body weight, dry matter intake (DMI), feed conversion ratio (FCR), average daily gain (ADG) and carcass weight and dressing percentage. As for proximate meat analyses, there were no significant differences between treatments for crude protein, moisture, ether extract, ash, pH and colour parameters such as the luminosity index (L*), red index (a*) and yellow index (b*) (Table 3).

### 3.2. Fatty Acid Composition

The effects of dietary treatment on the individual FA are reported in Table 4. The total of saturated fatty acids (SFA) was not different between groups. Within the SFA class, the dietary treatment with 20% and 40% olive cake increased the levels of C12:0 (*p* < 0.001), C14:0 (*p* < 0.001) and C21:0 (*p* < 0.05) compared to the control group. Additionally, the C15:0 content (*p* < 0.05) decreased in meat samples from OC40 and OC20 treatments compared to the C group.

The total monounsaturated fatty acid (MUFA) was enhanced by olive cake inclusion (*p* < 0.001). In particular, the level of oleic acid (C18:1n-9) increased in the muscle of pigs fed with the silage olive cake inclusion (*p* < 0.001), although no difference was observed as the level of inclusion of silage olive cake in the diet increased from 20% to 40% (dry matter).

In relation to the monounsaturated fatty acids, the level of C16:1 n-9 (*p* < 0.001) fatty acid increased in OC20 and OC40 meat samples compared to those of C group; conversely, a higher proportion of C14:1 n-9 (*p* < 0.01) was observed in meat samples from C group compared to meat samples from OC20 and OC40 groups.

Feeding the ensiled olive cake diet decreased the total PUFA levels. Consequently, the polyunsaturated/saturated ratio (P/S) significantly changed (*p* < 0.001 and *p* < 0.01, respectively), although no differences were observed between OC20 and OC40 groups.

The levels of linoleic acid (C18:2n-6; *p* < 0.001) and γ-linolenic acid (C18:3n-6; *p* < 0.01) decreased in the OC20 and OC40 samples compared to those of the C group.

A higher proportion of C20:3n-6 (*p* < 0.001), C20:3n-3 (*p* < 0.001), C20:4n-6 (*p* < 0.001), C20:5n-3 (*p* < 0.001) and C22:6n-3 (*p* < 0.001) was noticed in muscles from C treatment compared to the OC20 and OC40 ones. Conversely, the amount of C20:2n-6 increased (*p* < 0.01) in the OC40 group more than in other groups.

The dietary treatment affected the sum of n-3 PUFA and n-6 PUFA, with a greater concentration found in meat samples from the C group (*p* < 0.01 and *p* < 0.001, respectively) compared to those in other groups.

No effect of dietary treatments was observed on the atherogenic (IA) and thrombogenic indexes (TI) and the ratio between hypocholesterolemic and hypercholesterolemic fatty acids (h/H index).

### 3.3. PCA Analysis

Figure 1 shows the PCA scores plots and the corresponding loadings for the PC1 and PC2, which together account for 88.81% of the total variance, respectively. The first component (PC1) explains 81.77%, and the second component (PC2) represents 7.03% of the total information.

The labels correspond to the following variables: MUFA, monounsaturated fatty acids; PUFA, polyunsaturated fatty acids; n-3 = (C18:3 n-3 + C20:2 n-3 + C20:3 n-3 + C20:5 n-3 + C22:5 n-3 + C22:6 n-3); n-6 = (C18:2 n-6 + C18:3 n-6 + C20:2 n-6 + C20:3 n-6 + C20:4 n-6 + C22:2 n-6 + C22:4 n-6).

## 4. Discussion

### 4.1. Animal Performances and Meat Quality Measurements

The performance parameters were not affected by dietary treatment. No significant differences were observed in the final weight and carcass weight among the experimental groups (Table 2).

These results are in accordance with other studies in which the inclusion of olive cake in the diets was not shown to affect the growth of the animals [7,33]. Additionally, as reported by Aboagye et al. [34], the Apulo-Calabrese is a breed characterised by reduced growth and carcass performance, and this justifies the lower ADG values registered in all treatments of our trial.

As for proximate meat analyses, the finishing diet with silage olive cake did not significantly affect the pH and colour parameters of meat, according to Ferrer et al. [14].

### 4.2. Meat Fatty Acid Composition

The inclusion of fats and oils in monogastric animal feeding is a common practice due to their high energetic input and the availability of essential fatty acids, improving production. Additionally, dietary fat alters the lipid composition as well as the nutritive and organoleptic qualities of meat. Oleic acid (C18:1 n-9; 55–83%) is particularly abundant in silage olive cake, and the consumption of this fatty acid has been strongly associated with health benefits [35,36].

As expected, the FA profile of intramuscular fat changed as a result of the inclusion of ensiled olive cake in the diet, with a higher proportion of MUFA and a lower proportion of PUFA. The high oleic acid content of olive cake dietary treatments is linked to the increase in MUFA detected in fat tissue.

These results agree with the findings of Ferrer et al. [14], who found a higher concentration of MUFA and a lower content of PUFA in pigs when olive cake (in the range of 0–15%) was included in the diet, reflecting the dietary fatty acid composition. Additionally, the modification of the FA profile with the addition of olive by-products has been described by González et al. [37] and Serra et al. [38] as being of interest due to the improved sensory quality of meat.

Considering the PUFA content, in our study, the values ranged from 28.62% in the control group to 15.35% in the OC40 group, fed with the inclusion of 40% of silage olive cake. Some interesting differences are detectable when compared to Liotta et al. [8], who found higher levels of PUFA in Pietrain pigs fed with the inclusion of olive cakes (6 to 7% more) in the diet.

The amount and type of dietary fat, the de novo synthesis of fatty acids, the conversion rate to other fatty acids and metabolites, the percentage of oxidation for energy consumption and the levels of dietary elements and vitamins affect the amount of PUFAs in any tissue [14,31]. In our study, the lower proportion of PUFA detected in the muscle of pigs fed with the inclusion of olive cake is likely to be associated with the lower linoleic acid contents provided by the OC20 and OC40 dietary treatments (see Table 1). Furthermore, it should be noted that the PUFA percentage in the OC 40 group has been found within the limit of 15%, which is considered the threshold above which the fat consistency and oxidative stability could be negatively affected [39].

Nutritional evaluation of the fat fractions of foods is frequently based on nutritional indexes such as the polyunsaturated/saturated (P/S) ratio and n-6/n-3 ratio [40], and the atherogenicity (AI), thrombogenicity (TI) and hypocholesterolemic/hypercholesterolemic (h/H) characteristics [40,41].

The inclusion of olive cake in the pig-fattening diet improves the P/S ratio of meat toward the required value of 0.40 in human nutrition. The P/S ratio found in meat from the OC 20 and OC 40 groups is comparable to the value reported by Oksbjerg et al. [42] in pigs raised under organic and free-range production systems and over the value of 0.16 found by Leite et al. [18] in Bisaro pigs fed with olive cake supplementation.

The n-6/n-3 PUFA ratio was not significant between treatments. However, the values were high in relation to the recommended value of 4, which is associated with benefits to human health [43].

As stated by several authors [44,45], it is difficult to decrease the n6/n3 ratio in pork due to the high concentration of C18:2n-6 found in typical feed concentrate. In our study, all dietary treatments were richer in C18:2n-6 (see Table 1); thus, it is not surprising to find a high n-6/n-3 PUFA ratio in the meat from all experimental groups. Regarding the n6/n3 ratio, a similar value was found in the Nero Siciliano breed by Zumpo et al. [45] and in pigs fed under free-range conditions by Oksbjerg et al. [42].

The atherogenic (IA) and thrombogenic indexes (TI) are measures of lipid quality, which could serve as predictors of cardiovascular risks [41]. They were lower than 1.00. Dietary treatments did not significantly affect the IA and IT indices on the intramuscular fat of muscle, and the atherogenicity potential was within the expected range. The IA and IT values agreed with those achieved by Liotta et al. [8] in Pietrain pigs fed with 50 g/kg and 100 g/kg of olive cake incorporated into their diet. 

The hypo-/hypercholesterol (h/H) ratio is an index that considers the functional activity of FAs in the metabolism of plasma cholesterol lipoproteins, whose types and quantities are associated with a higher or lower risk of cardiovascular disease. Nutritionally higher h/H values are considered more beneficial for human health. The average of h/H (2.97) is high in all experimental groups compared to the values reported by Woloszyn et al. [46] in pork (2.4); this confirms that meat from Apulo-Calabrese pigs reared in a free-range system with the supplementation of silage olive cake, is recommended for a healthy diet.

### 4.3. PCA Analysis

The standard principal component analysis (PCA) technique was applied as a tool for reducing data dimensions to a few principal components and visualising similarities. It produces a new set of variables derived from the best linear combination of the original parameters, which accounts for more variance than any other combination. Simultaneously, this chemometric tool allows the identification of how the samples differ and which variables contribute the most to the difference. Considering the score plots for PC1 and PC2, the difference among the meat samples from different dietary treatments is apparent (Figure 1).

The meat samples from pig feed with diet C are located in the lower right of the plot (PC1 negative and PC2 positive) and can be clearly separated from all others, whereas the meat samples from pig feed with the inclusion of silage olive cake at 20% and 40% of dry matter in the diet (OC20 and OC40), clustered together on the inferior (PC1 and PC2 negative) and superior (PC1 positive and PC2 negative) left of the plot, can be partly distinguished. Moreover, as can be seen in Figure 1, the clear separation of meat samples, according to the presence or absence of the ensiled olive cake in the diets, is evident. The FAs accounting for most variation in the PCA were n-3 long-chain fatty acids, n-6 long-chain fatty acids, C18:2n-6, C20:3n-6, C20:3n-3, C20:4n-6, C20:5n-3, C22:6n-3, C14:1 ci-s9 and PUFA (positive loading), and C12:0, C14:0, C18:1n9 and MUFA (negative loading) for PC1. The second principal component (PC2) captured the variance associated with C16:1 (positive values) and C20:2n6 (negative values). From the comparison of the PCA score plot and PCA loading plot in Figure 1, it may be inferred that the contents of n-3 long-chain fatty acids, n-6 long-chain fatty acids, C18:2n-6, C20:3n-6, C20:3n-3, C20:4n-6, C20:5n-3, C22:6n-3, C14:1 ci-s9 and PUFA were higher in control samples (C), whereas meat samples from the OC20 and OC40 groups were characterised by high values of C12:0, C14:0, C18:1n-9, C16:1n-9, C20:2n-6 and total MUFA.

## 5. Conclusions

The main finding of this study was that including ensiled olive cakes in the fattening diet of Apulo-Calabrese pigs had no effect on animal performance and carcass characteristics. Moreover, the replacement of part of the concentrate with ensiled olive cake in the diet improves the fatty acid composition of meat from a health point of view, increasing oleic acid and decreasing linoleic acid and PUFA levels. Thus, from a technological and commercial point of view, the inclusion of silaged olive cake in fattening diets results in meat with a fatty acid profile more appropriate for transformation and storage. In fact, the food industry prefers meat with a limited amount of polyunsaturated fatty acids because they are more likely to undergo lipolytic and oxidative processes, resulting in rancidity, abnormal flavours, fat softness, and altered organoleptic properties.

This investigation may provide the basis for further studies, including the evaluation of lipid stability during the refrigerated storage of meat and the flavour of meat products, enabling broader application in the research and food industries.

Furthermore, the valorisation of local breeds, such as Apulo-Calabrese, would lead to the creation of a niche market with increasing importance and very high profits.

Finally, the incorporation of ensiled olive cake at up to 40% in diets for fattening pigs could represent a strategy in Mediterranean areas to naturally improve the nutritional value of meat and meat products as well as to valorise an olive industry by-product while reducing the environmental impact and promoting the exploitation of this local feed resource according to the principles of the circular economy.

## Figures and Tables

**Figure 1 animals-13-02022-f001:**
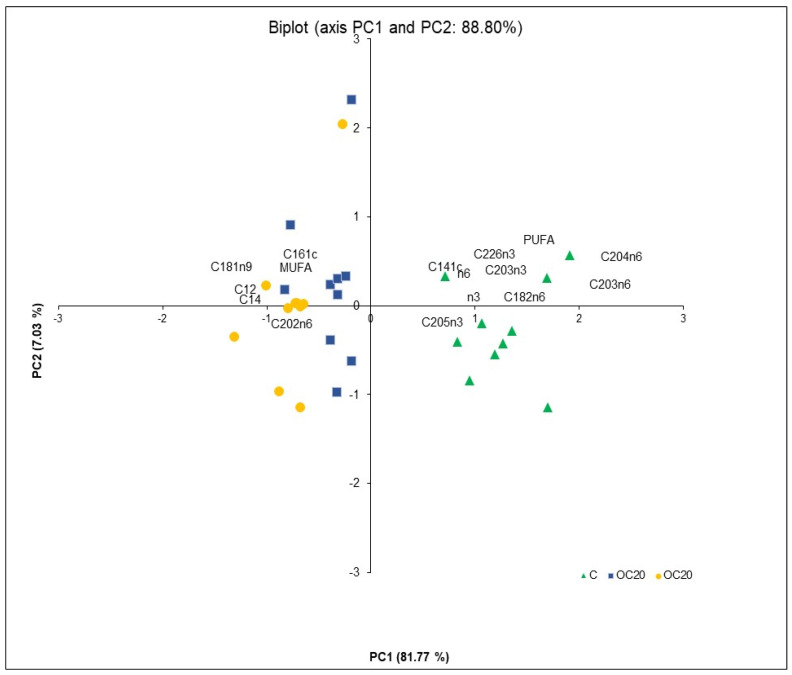
Principal component analysis was applied to fatty acids detected in meat samples from concentrate (C), concentrate with the inclusion of silage cake at 20% (OC20) and concentrate with the inclusion of silage cake at 40% (OC40) diets. Scores and loading plots on the first two principal components. Scores correspond to samples, and loadings correspond to variables.

**Table 1 animals-13-02022-t001:** Ingredients (g/100 g of DM) and chemical composition of diets offered.

	Diet ^1^		
	C	OC20	OC40
Barley grain	29.5	26.5	16.5
Maize grain	29.5	26.5	16.5
Wheat bran	20.0	-	-
Faba bean	20.0	26.0	26.0
Olive cake silage	-	20.0	40.0
Vitamin-mineral premix ^2^	1.0	1.0	1.0
Chemical composition			
Dry matter (DM), g/kg wet weight	901	810	713
Crude protein, g/kg DM	162	154	155
Ether extract, g/kg DM	27	40	54
Ash, g/kg DM	32	34	37
NDF, g/kg DM	255	270	341
ADF, g/kg DM	94	151	218
ADL, g/kg DM	15	33	56
Gross energy, MJ/kg DM	18.76	18.91	18.89
Fatty acids (g/100 g of fatty acids)			
C16:0	15.92	14.49	13.69
C18:0	2.02	2.09	2.23
C18:1 n-9	20.72	31.75	42.25
C18:2 n-6	56.37	47.45	37.48
C18:3 n-3	3.75	3.19	2.52
Others	1.22	1.03	1.83

^1^ C = Control diet; OC20 = diet with silage olive cake at 20% of dry matter; OC40 = diet with silage olive cake at 40% of dry matter. ^2^ Provided the following quantities per kilogram of diet: vitamin A, 6750 IU; vitamin D3, 1000 IU; vitamin E, 2.0 mg; vitamin B12, 0.01 mg; vitamin B1, 1.0 mg; folic acid 0.2 mg; D-pantothenic acid, 5.0 mg; cobalt, 0.05 mg; manganese, 12.5 mg; zinc, 15.0 mg; molybdenum 0.5 mg.

**Table 2 animals-13-02022-t002:** Pig performances in vivo and data at slaughter.

	Diets ^1^				
	C	OC20	OC40	S.E.M. ^5^	*p*-Value
Number of pigs	10	10	10		
Final body weight, kg	145.10	146.33	144.08	5.781	0.945
ADG ^2^ (g/d)	412.50	421.50	402.83	9.171	0.758
Total DMI ^3^, kg/d	3.35	3.07	2.98	0.203	0.546
FCR ^4^, g DMI/g ADG	8.12	7.52	7.40	0.567	0.439
Carcass weight, kg	120.08	121.83	119.16	8.330	0.751
Dressing percentage, %	82.68	83.24	82.71	1.021	0.683

^1^ C = Control diet, OC20 = diet with silage olive cake at 20% of dry matter, OC40 = diet with silage olive cake at 40% of dry matter; ^2^ ADG = average daily gain; ^3^ DMI = dry matter intake; ^4^ FCR = feed conversion ratio; ^5^ S.E.M. = standard error of the means.

**Table 3 animals-13-02022-t003:** Effect of the dietary treatment on chemical composition (g/100 g of muscle), pH and colour of *Longissimus thoracis* muscle.

	Diets ^1^				
	C	OC20	OC40	S.E.M. ^2^	*p*-Value
Moisture	74.083	74.013	73.199	0.387	0.232
Protein	20.663	21.149	20.928	0.223	0.334
Fat	1.860	1.905	2.239	0.407	0.774
pH_48_ ^3^	5.766	5.800	5.850	0.066	0.682
Lightness (L*)	49.316	49.203	48.277	1.864	0.967
Redness (a*)	3.905	3.965	6.081	0.868	0.164
Yellowness (b*)	5.854	6.198	7.249	0.715	0.296

^1^ C = Control diet, OC20 = diet with silage olive cake at 20% of dry matter, OC40 = diet with silage olive cake at 40% of dry matter; ^2^ S.E.M. = standard error of the least square means; ^3^ pH_48_ = pH at 48 h after slaughter.

**Table 4 animals-13-02022-t004:** Effect of the dietary treatment on fatty acid composition (mg/100 g of meat) of *Longissimus thoracis* muscle.

	Diets ^1^				
	C	OC20	OC40	S.E.M. ^2^	*p*-Value
**C10:0**	1.230	1.527	1.776	0.601	0.068
**C12:0**	1.019 ^B^	1.281 ^A^	1.325 ^A^	0.057	0.0008
**C14:0**	17.153 ^B^	23.007 ^A^	23.908 ^A^	0.877	0.0001
**C14:1n-9**	0.509 ^d^	0.315 ^e^	0.282 ^e^	0.050	0.012
**C15:0**	1.254 ^a^	1.004 ^b^	0.833 ^b^	0.092	0.019
**C16:0**	400.47	427.16	427.35	17.52	0.562
**C16:1 t**	5.372	6.021	5.564	0.363	0.304
**C16:1n-9**	57.987 ^B^	72.937 ^A^	72.027 ^A^	2.260	0.0004
**C17:0**	5.423	4.877	4.382	0.482	0.272
**C17:1**	4.136	4.493	3.702	0.536	0.579
**C18:0**	207.96	203.21	195.622	8.958	0.238
**C18:1t11**	3.288	3.167	3.863	0.497	0.466
**C18:1 n-9**	682.355 ^B^	844.561 ^A^	921.361 ^A^	23.283	0.0001
**C18:1 n-7**	75.122	82.937	85.988	2.816	0.057
**C18:2t9t12**	2.798 ^b^	3.363 ^ab^	3.501 ^a^	0.191	0.047
**C18:2 n-6**	349.686 ^A^	244.785 ^B^	209.244 ^B^	14.587	0.0001
**C20:0**	1.819	2.212	2.351	0.165	0.097
**C18:3n-6**	2.144 ^d^	1.454 ^e^	1.202 ^e^	0.167	0.003
**C20:1**	1.380	1.321	1.365	0.194	0.869
**C18:3 n-3**	12.521	12,046	12.533	0.654	0.795
**C21:0**	4.347 ^b^	5.413 ^ab^	6.627 ^a^	0.580	0.044
**C20:2 n-6**	13.847 ^e^	17.960 ^de^	22.045 ^d^	1.422	0.004
**C22:0**	1.833	1.855	1.892	0.291	0.913
**C20:3 n-6**	13.162 ^A^	9.927 ^B^	9.653 ^B^	0.479	0.0002
**C20:3 n-3**	13.949 ^A^	7.960 ^B^	6.809 ^B^	1.177	0.0013
**C20:4 n-6**	132.431 ^A^	61.705 ^B^	41.953 ^B^	13.041	0.0005
**C20:5 n-3**	6.197 ^A^	4.649 ^B^	4.526 ^B^	0.253	0.0004
**C22:2 n-6**	2.094	1.261	2.145	0.699	0.614
**C22:4 n-6**	16.075	6.528	5.342	3.538	0.063
**C22:5 n-3**	18.682	14.282	8.658	3.012	0.093
**C22:6 n-3**	12.994 ^A^	6.624 ^B^	4.654 ^B^	1.314	0.0011
**n-3 ^3^**	63.293 ^d^	45.561 ^e^	37.183 ^e^	4.368	0.0019
**n-6 ^4^**	516.592 ^A^	325.662 ^B^	269.539 ^B^	27.928	0.0001
**n-6/n-3**	8.161	7.147	7.249	0.388	0.309
**SFA ^5^**	642.508	671.547	666.066	25.662	0.842
**MUFA ^6^**	830.149 ^B^	1015.552 ^A^	1094.152 ^A^	25.261	0.0001
**PUFA ^7^**	596.530 ^A^	392.544 ^B^	332.265 ^B^	30.609	0.0001
**P/S ^8^**	0.928 ^d^	0.584 ^e^	0.499 ^e^	0.079	0.0031
**AI ^9^**	0.333	0.375	0.374	0.019	0.263
**TI ^10^**	0.746	0.826	0.826	0.047	0.409
**h/H ^11^**	3.163	2.864	2.892	0.209	0.646

Means in the same row with different letters indicate significant differences (a, b = *p* < 0.05; d, e, = *p* < 0.01; A, B = *p* < 0.001). ^1^ C = Control diet, OC20 = diet with silage olive cake at 20% of dry matter, OC40 = diet with silage olive cake at 40% of dry matter. ^2^ SEM, standard error of means; ^3^ n-3 = (C18:3 n-3 + C20:2 n-3 + C20:3 n-3 + C20:5 n-3 + C22:5 n-3 + C22:6 n-3); ^4^ n-6 = (C18:2 n-6 + C18:3 ^4^ n-6 + C20:2 n-6 + C20:3 n-6 + C20:4 n-6 + C22:2 n-6 + C22:4 n-6); ^5^ SFA, saturated fatty acids; ^6^ MUFA, monounsaturated fatty acids; ^7^ PUFA, polyunsaturated fatty acids; ^8^ P/S polyunsaturated fatty acids/saturated fatty acids; ^9^ AI, atherogenic index = C12:0 + 4 × C14:0 + C16:0)/(n-3 + n-6 + MUFA); ^10^ TI, thrombogenic index = (C14: 0 + C16: 0 + C18:0)/((0.5 × MUFA) + (0.5 × n-6)+ (3 × n-3) + (n-3/n-6); ^11^ h/H= [(C18:1n 9, C18:1n 7, C18:2n 6, C18:3n 6, C18:3n 3, C20:3n 6, C20:4n 6, C20:5n 3, C22:4n 6, C22:5n 3 and C22:6n 3)/(C14:0 and C16:0)].

## Data Availability

The authors confirm that the data supporting the findings of this study are available on request.

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
