# Peer review of "Effect of Dietary Ensiled Olive Cake Supplementation on Performance and Meat Quality of Apulo-Calabrese Pigs"

_animals, 2023, doi:10.3390/ani13122022_

Round 1

Reviewer 1 Report

In general, the Manuscript is well structured, data presented and discussed in a good manner. However, the study could be improved with the oxidative stability of feed ingredient (storage during 120 day?) as well as pork meat during the chilled storage (if this is possible to be examined?). Therefore, the suggestion is that authors provide more information (oxidative stability of pork) about quality properties of pork from the feeding trial.    

Specific comments:

Line 28-29: missing information about the feeding period, thus, inclusion level per dry matter should be added.

Line 57: be consistent through the text, please see line 56

Line 99-100: please, revise or make this clear, as now it sounds that in a fattening phase pigs are only indoor fed. is that correct?

Table 1: Elaborate differences in ash and "other FA" in OC20.

Section 2.4. does not explain if pigs were slaughtered at the same day? Also, no information how the loin was packed and stored for pH and color analysis.

Section 2.5. After how many days post mortem color measurements were done? Have the samples been during the whole period exposed to air or packed?

Line 190: Elaborate why GLM analysis were used? Has a model been tested, if yes, what were the factors and their interaction?

Table 2 and 3: its not clear why authors used "a, b: P<0.05; A, B: P<0.01" when there were not significant differences between quality traits.

Table 4 needs to be adjusted to the rest of the tables in the Manuscript.

 Discussion could be improved.

Line 316-317: Please, revise it, as it is not clear what authors wanted to say.

Line 331-334: Very confusing and long sentence, please revise.

Line 335-344: Please, revise so its clear what you are discussing here.

English language requires improvements. 

Reviewer 2 Report

The subjects of  the manuscript meet the objectives of  ANIMALS.  I have advised that the manuscript will be reconsidered for publication after minor revision.

   However, there are a high number of remarks that should be addressed before the completely new re-submission of the manuscript. Required changes or suggestions are listed below:

1. INTRODUCTION  

The weakness of the Introduction section is the very poorly emphasized novelty of these studies;

The hypothesis of Authors what kind of results they expected is not given; 

Lines 68-74: The bioaccumulation of fatty acids (derived from dietary oils) in animal tissues dependent upon concentrations of elements and vitamins in animals' diets; in fact, dietary trace elements (like selenium, zinc, iron, etc.) afect pancreatic secretion of lipase. Thus, please cite the publication: Białek M., Karpińska M., Czauderna M., 2022. Enrichment of lamb rations with carnosic acid and seleno-compounds affects the content of selected lipids and tocopherols in the pancreas. J. Anim. Feed Sci. 31(2), 161-174. https://doi.org/10.22358/jafs/147089/2022

Moreover, please provide concentrations of elements/composition of mineral and vitamin mixture (premix) in the control and experimental diets (Table 1);

2.  Materials & Methods

Table 1;  please provide "gross energy" for the control and experimental diets;

Lines 179 - 188  The "Fatty acid analysis" section: 

Line 181:  Please replace   "Fat"  with "Fatty acids in fat were methylated ......  ";

- please add volume(s) of injected samples containing methylated fatty acids (FAME);

- please specify a split ratio, temperature of an injector, used the carrier                gas and the length of the capillary GC-column; 

-  the carrier gas operated at a constant pressure (so, please specify    values of a pressure and initial flow rate) ? or at a constant flow        rate (so, please specify a value of flow rate) ?

- a fatty acid composition in analysed L.T. muscles depends upon the selectivity of used gas-chromatography-FID method; so, please specify: the column temperature programm;

3. RESULTS

Table 4:  Please specify the concentration of fatty acids (FAs) in Longissimus thoracis muscle  as  mg of fatty acids/100 g of Longissimus thoracis muscle.  The concentration of fatty acids in the muscle presented as mg of FAs/100 g of Longissimus thoracis muscle  clearly determines the influence of experimental diets the development of health-promoting values of Longissimus thoracis muscle.

The authors use strange, unusable values of fatty acids concentrations in muscles:  "fatty acid composition (g/100g of the total methyl esters) of Longissimus thoracis muscle";

Table 4:  please add: the hypocholesterolemic/hypercholesterolemic FA (h/H-Ch) ratio in Longissimus thoracis muscle (see Table 5 in the publication: publication: BiaÅ‚ek M., KarpiÅ„ska M., Czauderna M., 2022. Enrichment of lamb rations with carnosic acid and seleno-compounds affects the content of selected lipids and tocopherols in the pancreas. J. Anim. Feed Sci. 31(2), 161-174. https://doi.org/10.22358/jafs/147089/2022 ).  

CONCLUSION:

The authors should provide more extensive justification of the final conclusion;

Furthermore, express please new implications for the further scientific studies. 

Minor editing of English language required.

Round 2

Reviewer 1 Report

Dear Authors,

thank you for revised Manuscript and changes done to improve it. However, there are few comments I would like you to elaborate (green highlighted).
